# Asymmetric and Reduced Xanthene Fluorophores: Synthesis, Photochemical Properties, and Application to Activatable Fluorescent Probes for Detection of Nitroreductase

**DOI:** 10.3390/molecules24173206

**Published:** 2019-09-03

**Authors:** Kunal N. More, Tae-Hwan Lim, Julie Kang, Hwayoung Yun, Sung-Tae Yee, Dong-Jo Chang

**Affiliations:** 1College of Pharmacy and Research Institute of Life and Pharmaceutical Sciences, Sunchon National University, Suncheon 57922, Korea (K.N.M.) (T.H.L.) (J.K.) (S.T.Y.); 2College of Pharmacy, Pusan National University, Busan 46241, Korea

**Keywords:** fluorescence, xanthene fluorophore, reduced rhodafluor, activatable fluorescent probe, nitroreductase

## Abstract

Xanthene fluorophores, including fluorescein, rhodol, and rhodamines, are representative classes of fluorescent probes that have been applied in the detection and visualization of biomolecules. “Turn on” activatable fluorescent probes, that can be turned on in response to enzymatic reactions, have been developed and prepared to reduce the high background signal of “always-on” fluorescent probes. However, the development of activity-based fluorescent probes for biological applications, using simple xanthene dyes, is hampered by their inefficient synthetic methods and the difficulty of chemical modifications. We have, thus, developed a highly efficient, versatile synthetic route to developing chemically more stable reduced xanthene fluorophores, based on fluorescein, rhodol, and rhodamine via continuous Pd-catalyzed cross-coupling. Their fluorescent nature was evaluated by monitoring fluorescence with variation in the concentration, pH, and solvent. As an application to activatable fluorescent probe, nitroreductase (NTR)-responsive fluorescent probes were also developed using the reduced xanthene fluorophores, and their fluorogenic properties were evaluated.

## 1. Introduction

The xanthene scaffold-based fluorophores, including fluorescein, rhodamine, and their hybrid structure (rhodol), are among the most commonly used fluorophores, which have been widely applied as fluorescent probes to detect various biological and cellular processes [1,2]. Rhodol and rhodamine fluorophores in the form of activity-based fluorescent probes have attracted much interest, due to their high quantum yields in aqueous solutions, over a broad pH range and better photostability, as compared to fluorescein in detecting various cellular phenomena [3]. Previously, fluorescent probes carrying macromolecules or small-molecule ligands for targeted delivery have been developed and used in biomedical imaging. These fluorescent probes are disadvantageous, due to their “always-on” nature for imaging which lead to poor target-to-background ratios, that result from non-specific binding. Activatable “turn-on” fluorescent probes were assisted in overcoming this drawback, as they emit fluorescence only under specific conditions, such as binding to the target protein [1,4,5]. Our research group has attempted the development of nitroreductase-responsive fluorescent probes for hypoxia imaging, based on xanthene fluorophores, such as fluorescein and rhodol fluorophores, which bear a carboxylate group [6]. However, the selective functionalization of the terminal -OH or -NH_2_ group on the xanthene ring via alkylation or amide coupling is difficult, because carboxylate is a good leaving group (Figure 1A). Alkylation or amide coupling of typical xanthene fluorophores mainly produce an undesired product via a reaction of the carboxylate group in its open form, and leads to low yield of the desired product. We envisioned that alkylation or amide coupling could be achieved at the desired position by shifting the equilibrium toward closed form (Figure 1B). Reduced xanthene fluorophores have a benzyl alkoxy group, which is a poorer leaving group, compared to the carboxylate group in typical xanthene fluorophores. Koide’s group first reported a reduced fluorescein derivative, containing a benzyl alcohol, instead of benzoic acid, producing Pittsburgh green, and it was capable of detecting palladium and mercury [7,8]. Urano’s group also reported various reduced xanthene fluorophores, based on rhodol and rhodamine containing benzyl alcohol (HMDER), benzyl thiol, and benzyl amine instead of the benzoic acid, and designed a novel activatable fluorescent probe, using HMDER for in vitro and in vivo imaging of β-galactosidase [9,10,11,12]. Subsequently, a few activatable fluorescent probes, based on reduced rhodafluors for biomedical imaging or sensing ozone, have been reported [13,14,15,16,17,18,19,20,21,22].

Asymmetric and reduced xanthene fluorophores based on fluorescein can be synthesized from commercially available fluorescein via the reduction and subsequent oxidation of the carboxylate group of fluorescein (Figure 2A), whereas reduced rhodol and rhodamine fluorophores can be prepared by a convergent synthesis of the xanthene moiety via the reaction of 2-(4-(dialkylamino)-2-hydroxybenzoyl)benzoic acid with resorcinol or 4-aminophenol, followed by reduction and subsequent oxidation (Figure 2B,C) [9,10,11,14,23]. However, the current synthetic methods have limitations in effectively generating diverse asymmetric and reduced xanthene fluorophores. Not only different types of fluorophores, such as fluorescein, rhodol, and rhodamine, but also the same type of fluorophores with different substituents should be individually synthesized. Thus, we have designed a diversity-oriented strategy, that is based on continuous Pd-catalyzed cross-coupling reactions, using fluorescein as the starting material for the efficient synthesis of a series of asymmetric and reduced xanthene fluorophores (Figure 2D).

Herein, we describe the synthesis and photochemical properties of a series of asymmetric and reduced xanthene fluorophores with fluorescein, rhodol, and rhodamine scaffolds, that enable efficient asymmetric *N*-functionalization or *O*-functionalization. Nitroreductase (NTR)-responsive fluorescent probes were developed as activatable fluorescent probes using the various reduced xanthene fluorophores, which can be applied to the design of new pro-drugs for therapeutics and imaging agents targeting hypoxia [6,23].

## 2. Results and Discussion

### 2.1. Chemistry

We started the synthesis of a series of asymmetric and reduced xanthene fluorophores from commercially available fluorescein, in order to produce fluorophores with synthetic advantage in *O*- and *N*-functionalization of the xanthene ring, and can be applied as an activatable fluorescent probes. The asymmetric and reduced fluorescein derivative (**3**) for *O*-functionalization was first prepared from fluorescein via dimethylation, LiAlH_4_ reduction, and subsequent oxidation with *p*-chloranil (Scheme 1) [10]. We attempted to extend our synthetic strategy to various reduced rhodafluors, including rhodol and rhodamine fluorophores, starting from **3**. Asymmetric and reduced rhodol derivatives with a monoalkylamino (*n*-propylamino, **7**), dialkylamino (diethylamino, **8**), or free amino (-NH_2_, **12**) group were synthesized from **3** via triflation and subsequent Pd-catalyzed cross-coupling reaction, which was developed by Tao Peng et al. [24,25,26,27].

The cross-coupling reaction, that has been used to prepare the reduced rhodol fluorophores was optimized by the reaction of triflate **5** with diethylamine, in the presence of conventional palladium catalysts (Pd(PPh_3_)_4_, Pd(OAc)_2_, and Pd_2_(dba)_3_·CHCl_3_), ligands (BINAP, Johnphos, and Xantphos), and base (Cs_2_CO_3_ and *t*-BuONa, data not shown). The use of Pd(OAc)_2_ (10 mol%), BINAP (16 mol%), and Cs_2_CO_3_ (3 eq.) in toluene at 105 °C led to the formation of reduced rhodol derivative **8** bearing a diethylamino group in moderate yield (41%). Under the same condition for the synthesis of **8**, cross-coupling reactions of **5** with *n*-propylamine and benzophenone imine produced reduced rhodol **7,** with an *n*-propylamino group in 20% yield and fluorophore **9,** which was further hydrolyzed, using 1 N HCl in THF to give reduced rhodol derivative **12** with an -NH_2_ group (55% yield over 2 steps). The reactions for reduced rhodol **7** and **8** required an excess of *n*-propylamine (20 eq.) and diethylamine (10 eq.), due to their low boiling points, which resulted in relatively low yields.

Next, we attempted to synthesize reduced rhodol **13** and **14** with a -OH group, which are key intermediates for the reduced rhodamine **17** and **18** with an -NH_2_ group. We designed a synthetic route, using intermediate **4,** bearing a methoxymethyl (MOM)-protected hydroxyl group, instead of the methoxy group (Scheme 1). Triflate **6** was synthesized in the same manner as **5**, except that the first methylation was replaced by MOM protection using MOMCl. Based on the cross-coupling reaction of **5**, Pd(OAc)_2_ and BINAP were used to synthesize the *O*-MOM protected rhodol fluorophores, containing an ethylamino (**10**) and diethylamino (**11**) group, but the reactions were unsuccessful. Thus, the Pd-catalyzed cross-coupling reaction of **6** for reduced rhodols (**10** and **11**) was optimized using a variety of Pd catalysts (Pd(OAc)_2_, Pd(PPh_3_)_4_, Pd(dppf)Cl_2_, and Pd_2_(dba)_3_·CHCl_3_) and ligands (BINAP, Xantphos, and Johnphos) (data not shown). Thus, **10** was synthesized in good yield (quant.) under the reaction conditions with Pd_2_(dba)_3_·CHCl_3_ (10 mol%), Xantphos (15 mol%), and Cs_2_CO_3_ (3 eq.) in toluene at 105 °C, while **11** was prepared in a moderate yield (47%) by using Pd(PPh_3_)_4_ (10 mol%), BINAP (15 mol%) and Cs_2_CO_3_ (3 eq.). *O*-MOM protected rhodol derivatives (**10** and **11**) were treated with trifluoroacetic acid to give rhodol fluorophores **13** and **14** bearing a phenolic -OH group for *O*-functionalization. Subsequently, we tried to synthesize reduced rhodamine fluorophores for *N*-functionalization. Triflation of rhodol derivative **13,** using triflic anhydride, gave the desired triflate **15** in very low yield. However, triflation by phenyl triflimide [*N*-phenyl-bis(trifluoromethanesulfonimide)] and K_2_CO_3_ in acetonitrile afforded the desired rhodol triflates **15** and **16** in moderate yields (69%, and 57%, respectively). We then performed Pd-catalyzed cross-coupling reactions of **15** and **16** to prepare reduced rhodamine fluorophores possessing an -NH_2_ group for *N*-functionalization. The cross-coupling reaction of **15** and **16** with benzophenone imine, using Pd(OAc)_2_, BINAP, and Cs_2_CO_3_ in toluene, followed by acidic hydrolysis produced the desired rhodamine fluorophores **17** and **18** in 44%, and 30% yields, respectively. Previously, various attempts were made for the synthesis of reduced xanthene fluorophores, which resulted in the synthesis of various fluorescein, rhodol, and rhodamine fluorophores from more than one synthetic scheme [9,10,11,14,23]. The continuous Pd-catalyzed cross-coupling reactions enabled the rapid synthesis of novel series of reduced fluorescein, rhodol, and rhodamine fluorophores in highly efficient and concise synthetic scheme. The library of chemically-stable reduced fluorescein, rhodols, and rhodamines could be constructed with this synthetic strategy in moderate- to high-yields. It can be applied to synthesize various reduced xanthene-based fluorophores with *N*-functionalization or *O*-functionalization, which are useful to the design and discovery of novel activity-based fluorescent probes.

Next, we investigated the reactivity of the reduced xanthene fluorophores in comparison with typical xanthene fluorophores for the *O*-alkylation and amide coupling reactions. We performed *O*-alkylation of reduced fluorescein **3** and typical fluorescein **22,** using methyl iodide and benzyl bromide, under basic conditions (Scheme 2). Both methylation and benzylation of **3** produced the desired *O*-alkylated product in high yields (86%, and 98%, respectively), whereas **22** only gave undesired ester products, which could not be used as activatable fluorescent probes. But the amide coupling reactions of two types of rhodol fluorophores (reduced rhodol **12** and typical rhodol **23**) were different from the *O*-alkylation reactions of **3** and **22**. In the amide coupling reactions, both **12** and **23** afforded the desired amide products (**21,** and **26**, respectively). However, under the reaction conditions, using HOBt and *i*PrNEt_2_ in DMF, EDC-coupling of **12** exhibited a six-fold higher yield (50%) than **23** (8%). From the results of alkylation and amide coupling of the two types of xanthene fluorophores, we concluded that reduced xanthene fluorophores are more beneficial for phenolic *O*-alkylation and terminal *N*-amide coupling, compared to typical xanthene fluorophores.

We further attempted to synthesize activatable fluorescent probes based on the reduced xanthene fluorophores for sensing nitroreductase, an enzyme that catalyzes the reduction of a nitro group to an amine via a hydroxyl amine in the presence of NADH and a representative biomarker of hypoxic cells including solid tumors [28,29] (Scheme 3). We envisioned that the reduced fluorescein (**3**), rhodol (**12** and **14**), and rhodamine (**18**) fluorophores will be potential candidates for activatable fluorescent probes, because they have a free -OH or NH_2_ group and show strong fluorescence at physiological pH (pH~7). The only exception is **12**, which shows weak fluorescence emission at pH~7, but we prepared an activatable fluorescent probe, based on **12** because the concentration of fluorophores released from activatable fluorescent probes can be calculated by the concentration-dependent calibration curve of the corresponding fluorophore. *O*-Alkylated NTR-responsive fluorescent probes (**27** and **28**), based on reduced fluorescein **2** and rhodol **14** containing an -OH group were synthesized using Ag_2_O in toluene, according to a previously reported method [6]. Reduced rhodol and rhodamine-based NTR-responsive fluorescent probes (**29** and **30**), bearing a carbamate linker, were prepared from reduced xanthene fluorophores **12** and **18** containing an -NH_2_ group, respectively, by using 4-nitrobenzyl chloroformate and *i*PrNEt_2_.

### 2.2. Photochemical Properties and NTR Reaction

The photochemical properties of the asymmetric and reduced xanthene fluorophores, including quantum yield, concentration-dependent fluorescence emission, stability, and solvent effect, were evaluated. The quantum yields of newly synthesized fluorophores were calculated in comparison with the reference standard, fluorescein (0.1 N NaOH, Φ_r_ = 0.85; Table 1). Most of reduced xanthene fluorophores exhibited significant quantum yields proving their promising fluorogenic nature. Reduced fluorophores with a -OH showed high quantum yield compared to fluorophore with an -NH_2_. Among all fluorophores, rhodol (**13**) and rhodamine (**17**) bearing monoethylamine showed highest quantum yields of 0.824, and 0.399 respectively, showing their strong fluorogenic nature.

Next, the fluorescent emission at the maximal absorption wavelength (λ_max_) for the asymmetric and reduced xanthene fluorophores in PBS (phosphate buffered saline; 10 mM, pH 7.4) was evaluated in a concentration-dependent manner (Figure 3). The reduced fluorescein (**3**), rhodol (**13** and **14**), and rhodamine (**17** and **18**), containing a -OH or -NH_2_ group, showed very strong fluorescence emission at physiological pH~7.4, whereas reduced rhodol **7** and **8,** without the -OH or -NH_2_ group, showed relatively weak fluorescence emission at pH~7.4. Reduced rhodol **8** with a -OCH_3_ and -NEt_2_ groups exhibited extremely low fluorescence and quantum yield, implying that an acidic hydrogen on the terminal oxygen or nitrogen atom on the xanthene ring is essential for sufficiently intense fluorescence emission in this series of fluorophores. The rhodol **12** showed exceptionally low fluorescent emission (about 2000 RFU at 5 μM) in the series, even though it contains an -NH_2_ group. Of all the reduced xanthene fluorophoresm containing the novel rhodol (**7**, **8** and **12**) and rhodamine (**17**) derivatives, the rhodamine **17** containing -NHEt and -NH_2_ groups showed the strongest fluorescence emission (about 80,000 RFU at 2 μM; Figure 3G).

However, the pH-dependent fluorescence spectra showed a different trend from physiological pH (Figure 4). Reduced rhodol fluorophores (**13** and **14**) showed strong fluorescence in the pH range of 5-11, indicating that it is feasible to apply them to activatable fluorescent probes for use under physiological conditions, as demonstrated previously [11,12]. On the other hand, the reduced fluorescein (**3**) and rhodamine (**17** and **18**) fluorophores exhibited, not only strong fluorescence at physiological pH, but also significantly enhanced fluorescence emission upon lowering the pH. Interestingly, reduced rhodol **7** and **12,** containing a -OCH_3_ at the end and mono-substituted amine (-NHPr), or free amine (-NH_2_), at the other end exhibited very weak fluorescence at physiological pH 7.4, but showed a significant increase in fluorescence below pH 6.

We then investigated the solvent effect on the fluorescence emission of the asymmetric and reduced xanthene fluorophores (Figure 5). It is well known that the fluorescence of xanthene dyes is complicated by the presence of a solvent-dependent equilibrium, between the colored open form bearing a zwitterion, and the colorless closed lactone form in protic solvents [30,31,32]. Urano’s group reported a series of reduced rhodol and rhodamine fluorophores, and concluded that the lifetime of the open form of reduced xanthene dyes is very important in determining fluorescence emission [11]. In addition to their results, we envisioned that the nucleophilicity of the benzylic alkoxide will shift the equilibrium towards non-fluorescent closed form, as the spirocyclization of the open form is induced by the nucleophilic addition of benzyl alkoxide (Figure 1B). As observed in many S_N_2 and nucleophilic addition reactions, polar protic solvents, such as water and methanol, can stabilize the nucleophile via solvation by hydrogen bonding and decrease its reactivity, whereas polar aprotic solvents, such as DMSO (dimethyl sulfoxide), which has a strong dipole moment, but cannot form H-bonds, enhance the reactivity of the nucleophile. Thus, we investigated the solvent effect on the fluorescence emission of reduced xanthene dyes, in order to evaluate the nucleophilicity of benzyl alkoxide in equilibrium between the closed and open forms (Figure 5). We measured the fluorescence emission of our reduced xanthene fluorophores (1.0 μM) in various solvents, including water (ε_r_ = 78.5 at 25 °C), methanol (ε_r_ = 32.6 at 25 °C), ethanol (ε_r_ = 24.6 at 25 °C), isopropanol (IPA, ε_r_ = 18.3 at 25 °C), and DMSO (ε_r_ = 47.0 at 25 °C). Most of the reduced xanthene dyes showed the strongest fluorescence in water and the weakest fluorescence in DMSO. The trends in the fluorescence emission, observed in polar protic solvents, were proportional to the dipole moment of the solvent, except for reduced rhodol fluorophore **14**. The fluorescence emission in polar protic solvents decreased with the decreasing dipole moment of the solvent. Whereas, a polar aprotic solvent DMSO, with a dielectric constant between those of water and methanol, showed lesser fluorescence emission, compared to the polar protic solvent for all the reduced xanthene fluorophores. The solvent effect on fluorescence emission implies that the nucleophilicity of benzyl alkoxide in the reduced xanthene fluorophores affects the equilibrium between the fluorescent open form and non-fluorescent closed form. However, the structure-based fluorescence emission of xanthene dyes in equilibrium is very complicated; for example, xanthene dyes exist in several neutral and ionic forms in solution and their equilibria is susceptible to a variety of factors, including concentration, pH, temperature, and solvent, etc. [30,31,33,34,35,36].

Finally, we applied our asymmetric and reduced xanthene fluorophores as activatable fluorescent probes targeting nitroreductase. We chose four representative fluorophores, including fluorescein **3**, rhodols **12** and **14**, and rhodamine **18**, to develop NTR-responsive fluorescent probes. Reduced rhodol **12** showed weaker fluorescence as compared to the other fluorophores, employed as NTR-responsive probes. Nevertheless, we chose **12** as a fluorophore for activatable fluorescent probe, as the concentration-dependent fluorescence calibration curve could be established, so that the concentration of the fluorophore, released from the NTR-responsive fluorescent probe, could be determined. As reported in our previous work on NTR-responsive fluorescent probes [6], xanthene fluorophores are linked to the *p*-nitrobenzyl group via an ether (**27** and **28**) or carbamate (**29** and **30**) moiety, and reduction of the nitro group in the probe triggers the release of the fluorophore via 1,6-rearrangement elimination of the *p*-aminobenzyl group, leading to a turn-on fluorescent response.

The stability of the linkage in the NTR-responsive fluorescent probes (**27–30**) was assessed by estimating the fluorescence emission under varying temperature and pH conditions (Figure 6). All the probes showed weak, but stable, fluorescence emissions in the temperature range 25 to 45 °C and pH range 5–13, implying that they can be applied as activatable fluorescent probes for sensing a specific protein, NTR, under physiological conditions. Exceptionally, the strong fluorescence emission of reduced rhodamine-based probe **30,** containing a carbamate linkage, present in acidic pH range between 2 to 4, is the result of the fluorophore via hydrolysis under acidic conditions, which is highly correlated with the pH-dependent fluorescence spectrum of fluorophore **18** (Figure 4H).

We performed kinetic studies (1.0 μM of probes) of probes during the NTR reaction, using 10 μg/mL of this protein as a function of time, in order to investigate the activatable response of probes to NTR (Figure 7). Probe **27** and **28**, which are fluorescein- and rhodol-based activatable fluorescent probes, containing an ether linkage, showed strong fluorescence responses over time in the presence of NTR (Figure 7A). On the other hand, probes **29** and **30,** bearing a carbamate linkage, which were prepared via *N*-functionalization of the reduced rhodol and rhodamine fluorophores containing an -NH_2_ group, showed relatively weak fluorescence emission over time during the NTR reaction. 

We determined the concentration of the released fluorophores in the NTR reaction using the concentration-dependent calibration curves obtained for the fluorophores (Figure 7B–F). Distinctly different results were obtained from the calibration curves: Probes **27** and **29** showed the completion of the NTR reaction with almost 1.0 μM of the released fluorophore (**3** and **12**), whereas the concentration of the fluorophore released from probes **28** and **30,** reached only 0.167 μM (**14**) and 0.065 μM (**18**) (Figure 7B). Although, probe **29** gave 100% yield for the NTR reaction, the corresponding fluorophore **12** could not be applied as an activatable imaging probe at physiological pH due to its photochemical nature, i.e., weak fluorescence emission at physiological pH. Rhodamine-based fluorophore **18** showed enough fluorescence at physiological pH, but the corresponding NTR-responsive fluorescent probe **30** exhibited a poor release of the fluorophore, during the NTR reaction, demonstrating that **18** is not a good candidate for an activatable fluorescent probe. The reduced fluorescein **3** and rhodol **14** fluorophores, bearing an -OH group showed strong fluorescence at physiological pH, and the corresponding NTR-responsive fluorescent probes (**27** and **28**), also emitted strong fluorescence during the NTR reaction. However, probe **28,** based on the reduced rhodol gave a relatively low yield (~17%) in the NTR reaction, as compared to probe **27**. Taken together, the NTR reaction revealed that **3** and **14** are promising candidates for activatable fluorescent probes, and among all the asymmetric and reduced fluorophores studied, fluorescein **3** is the best choice for an activatable fluorescent probe.

## 3. Materials and Methods

### 3.1. Materials and Instrumentation

All reagents and solvents were purchased from Sigma-Aldrich Chemical Co. (St. Louis, USA), Tokyo Chemical Industries (Tokyo, Japan), Daejung Chemicals (Siheung-si, Korea), and Alfa Aesar (Ward Hill, MA, USA) and used without any further purification. Anhydrous solvents were purchased from Sigma-Aldrich Chemical Co. (St. Louis, MO, USA), and all reactions were performed under nitrogen atmosphere. Silica gel (ZEOprep 60 40–63 μm, Zeochem, Louisville, KY, USA) was used for flash column chromatography, and silica gel plates (Kiesegel 60F_254_, Merck, Darmstadt, Germany) were used for thin-layer chromatography. ^1^H and ^13^C-NMR spectra were measured on a JEOL JNM-ECZ400s/L1 (400 MHz) spectrometer (Jeol, Tokyo, Japan), with CDCl_3_ or DMSO-*d*_6_ as the NMR solvent (Cambridge Isotope Laboratories, Tewksbury, MA, USA). Chemical shifts are expressed in parts per million (ppm), and the coupling constant *J* is reported in hertz (Hz). Chemical shifts (in ppm) in ^1^H-NMR are based on the chemical shift of tetramethylsilane (δ = 0 ppm) in CDCl_3_ as an internal standard. The chemical shifts in ^13^C-NMR are reported in ppm relative to the centerline of the triplet at 77.0 ppm observed for CDCl_3_ or 39.5 ppm for DMSO-*d*_6_. All in vitro enzyme assays were performed by recording the absorbance and emission using a Synergy™ H1 microplate reader from BioTek Instruments (Winooski, VT, USA). Nitroreductase from *Escherichia coli* and NADH were purchased from Sigma-Aldrich Chemical Co. The lyophilized nitroreductase powder was dissolved in deionized water, fractionated, and immediately stored at −80 °C.

### 3.2. General Synthetic Procedures

#### 3.2.1. General Procedure A: Alkylation

K_2_CO_3_ (2.5 eq.) was added to a solution of fluorescein (1.0 eq.) in DMF, and the reaction mixture stirred at rt for 1 h under N_2_ atmosphere. Methyl iodide or chloromethyl methyl ether (3.0 eq.) was added dropwise to the reaction mixture, using a syringe pump with the rate of 5 mL/1hr, and the reaction mixture stirred at rt for 3–12 h. After completion of the reaction, ice water was added to the reaction mixture and stirred at 0 °C for 30 min. The resulting yellow solid was filtered (in the case of the MOM protection reaction, extraction was performed using ethyl acetate) and washed with water to completely remove K_2_CO_3_. The resulting solid was dried or purified by column chromatography to afford the desired compound.

#### 3.2.2. General Procedure B: LiAlH_4_ Reduction and p-Chloranil Oxidation

To a solution of compound **1** or **2** (1.0 eq.) in anhydrous THF was added LiAlH_4_ (2.0 eq.) at 0 °C. The reaction mixture was stirred at 0 °C for 4 h. After completion of the reaction, sodium sulfate decahydrate (5.6 eq.) was added to the reaction mixture at 0 °C and then stirred at rt for 30 min. The reaction mixture was filtered through a short pad of Celite, which was washed with CH_2_Cl_2_. The filtrate was concentrated in vacuo, and the crude product used in the next reaction without further purification. The crude compound, obtained from the LiAlH_4_ reduction, was dissolved in MeOH, followed by the addition of p-chloranil (3.0 eq.), and stirred at rt for 2 h. The reaction mixture was filtered, and the filtrate concentrated in vacuo. The residue was purified by flash column chromatography on silica gel to give the desired product.

#### 3.2.3. General Procedure C: Triflation

To a solution of the phenol derivative (1.0 eq.) in anhydrous CH_2_Cl_2_ or CH_3_CN was added pyridine, or K_2_CO_3_ (4.0 eq.), respectively, and the reaction mixture stirred at 0 °C for 20 min. Triflic anhydride or N-phenyl-bis(trifluoromethanesulfonimide) (2.0 eq.) was slowly added to the reaction mixture over 30 min, and then, the mixture was allowed to warm to rt and stirred for 3 h. The reaction was quenched with water and extracted with CH_2_Cl_2._ The organic layer was washed with aqueous 1 N HCl solution or saturated NH_4_Cl aqueous solution, followed by water and brine. The organic layer was dried over Na_2_SO_4_, filtered and concentrated in vacuo. The residue was purified by flash column chromatography on silica gel to give the desired product.

#### 3.2.4. General Procedure D: Pd-Catalyzed Cross Coupling Reaction

All glassware was dried in an oven before use. To a solution of the corresponding triflate (1.0 eq.) and amine (10 or 20 eq.) or benzophenone imine (1.2 eq.) in anhydrous toluene were added Pd(OAc)_2_, Pd(PPh_3_)_4_, or Pd_2_(dba)_3_.CHCl_3_ (0.1 eq.), BINAP or Xantphos (0.16 eq.), and Cs_2_CO_3_ (3.0 eq.), and the reaction mixture was heated at 105 °C for 4–12 h under N_2_ atmosphere. After completion of the reaction, the mixture was filtered through a short pad of Celite and washed with CH_2_Cl_2_. The filtrate was concentrated in vacuo, and the residue was purified by flash column chromatography on silica gel to give the desired product.

#### 3.2.5. General Procedure E: MOM-Deprotection

To a solution of the corresponding MOM protected compound (100 mg) in anhydrous CH_2_Cl_2_ (1 mL) at 0 °C, a solution of trifluoroacetic acid in CH_2_Cl_2_ [1 mL, TFA:CH_2_Cl_2_ = 1:1 (*v*/*v*)] was slowly added dropwise at 0 °C. After the addition of TFA was complete, the reaction mixture was allowed to warm to rt and stirred at rt for 1 h. The reaction was quenched with 1 N NaOH solution and extracted with CH_2_Cl_2_. The organic layer was dried over Na_2_SO_4_, filtered, and concentrated in vacuo. The residue was purified by flash column chromatography on silica gel to give the desired product.

### 3.3. Determination of Fluorescence Quantum Yield

The quantum yield (Φ_s_) of the newly synthesized reduced xanthene fluorophores was determined by comparing the integrated area under the curve of the sample, excited at 490 nm, with the reference fluorophore. Fluorescein (Φ_r_ = 0.85, 0.1 N NaOH) is used as the reference fluorophore for determining the quantum yield of reduced xanthene fluorophore. Absorption spectra and fluorescence spectra of fluorophores were recorded on Synergy H1^TM^ microplate reader (BioTek Instruments, Winooski, VT, USA) and FluoroMate FS-2 fluorescence spectrometer (Scinco, Seoul, Korea), respectively. The absorbance values are kept below 0.1 in order to minimize re-absorption effects. Fluorescence quantum yield (Φ_s_) of reduced xanthene fluorophores was calculated by using following Equation (1):(1)$$\Phi s=\Phi r\times \left(\frac{\mathrm{Ar}}{\mathrm{As}}\right)\times \left(\frac{\mathrm{Fs}}{\mathrm{Fr}}\right)\times {\left(\frac{\eta s}{\eta r}\right)}^{2}$$
Φ_s_ = Quantum yield of sample; A_r_ = Absorbance of reference at excitation wavelength; A_s_ = Absorbance of sample at excitation wavelength; F_r_ = Integrated area under emission curve of reference; F_s_ = Integrated area under emission curve of sample; η_r_ = Refractive index of solvent of reference; η_s_ = Refractive index of solvent of sample; and Φ_r_ = Quantum yield of fluorescein.

### 3.4. Concentration-Dependent Fluorescence Study of Fluorophores

A concentration-dependent study was performed by incubating different concentration, such as 0.5, 1, 2, and 5 µM of the fluorophore in PBS (pH 7.4) at 25 °C, and recording the fluorescence spectra at each concentration in 96-well microplate using Synergy H1 reader.

### 3.5. Effect of pH on Fluorescence Intensity of Fluorophores

A pH-dependent fluorescence change of fluorophore was performed by incubating 1 µM of the probe in a range of pH buffer solutions (pH 2 to 13) at 25 °C, and recording the fluorescence at each pH in 96-well microplate, using Synergy^TM^ H1 (BioTek Instruments).

### 3.6. Solvent Effect on the Fluorescence Emission of Fluorophores

The effect of solvent polarity on the fluorescent intensity was measured by incubating 1 µM of the fluorophore in different solvents, including water, methanol, ethanol, isopropanol, and DMSO. The fluorescence was measured at the respective excitation wavelength of the fluorophore, in a 96-well microplate, using Synergy^TM^ H1 (BioTek Instruments).

### 3.7. In vitro Nitroreductase Assay

All spectroscopic readings were recorded on Synergy^TM^ H1 (BioTek Instruments) using a 96-well microplate. The NTR reaction was performed in a total volume of 200 µL with the addition of 100 µL of PBS (10 mM, pH 7.4), 10 µL of probe stock solution (20 µM in DMSO), 20 µL of NADH solution (5 mM in 0.01 M aq. NaOH), and an appropriate volume of NTR solution (10 µg/100 µL in distilled water). The final volume was adjusted to 200 µL using PBS. The plate was incubated at 37 °C for an appropriate length of time with continuous shaking, and the emission spectra were recorded at the respective wavelengths with respect to time to prepare kinetic graph.

### 3.8. pH and Thermal Stability of Fluorescent Probes

A temperature-dependent assay was performed by incubating 1 µM of the probe in PBS (pH 7.4) at different temperatures for 20 min, and recording the fluorescence at each temperature. A pH-dependent study was performed by incubating 1 µM of the probe in a range of pH buffer solutions (pH 2 to 13) at 25 °C and recording the fluorescence at each pH in 96-well microplate, using Synergy^TM^ H1 (BioTek Instruments).

All synthetic procedures, 1H-NMR, 13C-NMR and HRMS of all compounds can be seen in the Appendix A.

## 4. Conclusions

In conclusion, we developed a highly efficient and versatile synthetic route to a series of asymmetric and reduced xanthene fluorophores, including fluorescein, rhodol, and rhodamine derivatives, which are representative xanthene scaffold-based dyes, and employed them as activatable fluorescent probes for sensing nitroreductase. A variety of asymmetric and reduced xanthene fluorophores, bearing an -OH or -NH_2_ group, capable of *O*- or *N*-functionalization to prepare activatable fluorescent probes, were synthesized from commercially available fluorescein by continuous Pd-catalyzed cross-coupling reactions. Their photochemical properties, including quantum yields, fluorescence emission under various conditions (variation in concentrations, pH, and solvents) were characterized. Two fluorophores, including fluorescein (**3**) and rhodol (**14**) bearing an -OH group for *O*-functionalization, and two other fluorophores, including rhodol (**12**) and rhodamine (**18**), containing an -NH_2_ group for *N*-functionalization were subjected to develop NTR-responsive fluorescent probes. These probes exhibited turn-on fluorescence by releasing the fluorophore in the NTR reaction. This work demonstrates that the asymmetric and reduced xanthene fluorophores synthesized using our strategy are useful in developing activatable fluorescent probes for diverse biological applications under physiological conditions.

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
