# Peer review of "Asymmetric and Reduced Xanthene Fluorophores: Synthesis, Photochemical Properties, and Application to Activatable Fluorescent Probes for Detection of Nitroreductase"

_molecules, 2019, doi:10.3390/molecules24173206_

Round 1
Reviewer 1 Report
This manuscript describes the synthesis of novel spirocyclic xanthene-based fluorophores and studies on their application as activatable fluorescent probes for sensing nitroreductase.
This is a nice study with a significant amount of work in. The paper is carefully written and the procedures/results are clearly presented.
I just have a few very minor details that need to be addressed:
1) use consistently 'eq.' or 'eq' . in some cases the one is used and in others the other
2) same as 1) for 'rt' and 'RT'
3) line 135 - in (N‐phenyl‐bis(trifluoromethanesulfonimide)) change outer brackets to square brackets [N‐phenyl‐bis(trifluoromethanesulfonimide)]
4) line 343 - please include the addition rate for the addition with syringe pump
5) line 356 : p‐chloranil --> p-chloranil
6) line 362 : N‐phenyl‐bis(trifluoromethanesulfonimide) --> N‐phenyl‐bis(trifluoromethanesulfonimide)
7) line 378: "was added trifluoroacetic acid (1 mL) at 0 oC (v/v = 1:1), slowly dropwise" --> "trifluoroacetic acid (1 mL) was slowly added dropwise (v/v = 1:1) "
8) line 378 : I am not sure to what '(v/v = 1:1)' is referred to
9) in the supporting information compounds 1, 8 and 22 are known compounds. please include the relevant references along with some text if the data are in agreement with those previously reported
Author Response
Reviewer Report 1
- use consistently 'eq.' or 'eq' . in some cases, the one is used and in others the other
Answer: We have changed all words ‘eq’ to ‘eq.’.
- same as 1) for 'rt' and 'RT'
Answer: We have changed all words ‘RT’ to ‘rt’.
- line 135 - in (N‐phenyl‐bis(trifluoromethanesulfonimide)) change outer brackets to square brackets [N‐phenyl‐bis(trifluoromethanesulfonimide)]
Answer: We have made correction as pointed.
- line 343 - please include the addition rate for the addition with syringe pump
Answer: We have made correction as pointed.
- line 356 : p‐chloranil --> p-chloranil
Answer: We have changed wording ‘chloranil or p-chloranil’ to ‘p-chloranil’.
- line 362 : N‐phenyl‐bis(trifluoromethanesulfonimide) --> N‐phenyl‐bis(trifluoromethanesulfonimide)
Answer: We have made correction as pointed.
- line 378: "was added trifluoroacetic acid (1 mL) at 0 oC (v/v = 1:1), slowly dropwise" --> "trifluoroacetic acid (1 mL) was slowly added dropwise (v/v = 1:1) "
Answer: We have made correction as pointed as following; “a solution of trifluoroacetic acid in CH2Cl2 [1 mL, TFA:CH2Cl2 = 1:1 (v/v)] was slowly added dropwise”
- line 378 : I am not sure to what '(v/v = 1:1)' is referred to
Answer: (v/v = 1:1) is the proportion of TFA: CH2Cl2. We have corrected to make it clear as following; “a solution of trifluoroacetic acid in CH2Cl2 [1 mL, TFA:CH2Cl2 = 1:1 (v/v)] was slowly added dropwise”
- in the supporting information compounds 1, 8and 22 are known compounds. please include the relevant references along with some text if the data are in agreement with those previously reported
Answer: We have made correction in supplementary information as pointed. We had uploaded an old version of SI in the first submission. Thus, we submitted the latest version of SI with the correction according to your comments in this revision. The latest version of SI contains correct compound numbers and their structures for understanding of readers. Compound 8 and 22 were changed to compound 2 and 24, respectively.
Other corrections:
- We have made corrections in all reference. We have included DOI number and also made some corrections for some references. So please go with newly added references.
- We made some corrections in Table Table 1 uploaded in the first submission had no the explanatory notes b. We added the correct explanatory notes (a and b) in Table 1.
Reviewer 2 Report
The manuscript describes synthesis, photochemical properties of reduced xanthene fluorophores. Authors succeeded in developing efficient synthetic methods using cross-coupling and elucidating fluorescent nature of their derivatives. Moreover, their application to NTR-responsive fluorescent probes was developed. I think the manuscript is well-written and appropriate for publication in Molecules, however, a minor revision is needed.
1) The compound number in the manuscript is different from that in the supporting information. For example, while the compound 7 bearing propylamino group was shown in the manuscript, it was designated compound 4 in the experimental section. Check them again.
2) One concern raised by the reviewer is whether this cross-coupling method is applicable to NH3 in place of benzophenone imine or not. In view of atom economy, it would be better to use NH3, obviating a deprotection step.
Author Response
Reviewer Report 2
- The compound number in the manuscript is different from that in the supporting information. For example, while the compound 7 bearing propylamino group was shown in the manuscript, it was designated compound 4 in the experimental section. Check them again.
Answer: Thank you for pointing this. We had uploaded an old version of SI in the first submission. Thus, we submitted the latest version of SI with the correction according to your comments in this revision. The latest version of SI contains correct compound numbers and their structures for understanding of readers.
- One concern raised by the reviewer is whether this cross-coupling method is applicable to NH3 in place of benzophenone imine or not. In view of atom economy, it would be better to use NH3, obviating a deprotection step.
Answer: Thank you for pointing this. Cross-coupling with NH3 can be applied to synthesize compounds in our manuscript, and directly afford rhodol or rhodamine derivatives with an -NH2 group without hydrolysis step compared to use benzophenone imine. We didn’t perform such experiment yet, but we would like to try it for our further research to synthesize new rhodol or rhodamine derivatives in future.
Other corrections:
- We have made corrections in all reference. We have included DOI number and also made some corrections for some references. So please go with newly added references.
- We made some corrections in Table Table 1 uploaded in the first submission had no the explanatory notes b. We added the correct explanatory notes a and b in Table 1.